# Responses of Pea (*Pisum sativum* L.) to Single and Consortium Bio-Fertilizers in Clay and Newly Reclaimed Soils

**DOI:** 10.3390/plants12233931

**Published:** 2023-11-22

**Authors:** Ghada Abd-Elmonsef Mahmoud, Amany H. A. Abeed, Hassan H. A. Mostafa, Omaima Abdel Monsef

**Affiliations:** 1Botany and Microbiology Department, Faculty of Science, Assiut University, Assiut 71516, Egypt; dramany2015@aun.edu.eg; 2Central Laboratory of Organic Agriculture, Agricultural Research Center, Giza 12619, Egypt; hhhalim79@yahoo.com; 3Soil, Water and Environment Research Institute, Agricultural Research Center, Giza 12619, Egypt; omaimaabdelmonsef@yahoo.com

**Keywords:** plant growth-promoting, consortium, plant production, phosphate solubilizing, potassium solubilizing, nitrogen fixing

## Abstract

The huge development of climatic change highly affects our crop production and soil fertility. Also, the rise in the uncontrolled, excessive use of chemical fertilizers diminishes the soil prosperity and generates pollutants, threatening all environmental life forms, including us. Replacement of these chemical fertilizers with natural ones is becoming an inevitable environmental strategy. In our study, we evaluated the responses of *Pisum sativum* L. to the action of single species and consortiums of plant growth-promoting bacteria (*Azotobacter chroococcum*, *Bacillus megaterium*, and *Bacillus cerkularice*) in clay and new reclaimed soil types in terms of phenotype, yield components, and physiological and biochemical responses. Data analysis showed single or consortium microbial inoculation significantly increased the measured traits under clay and calcareous sandy soils compared to the control. Shoot physiological and biochemical activities, and seed biochemical activities were significantly enhanced with the inoculation of pea seeds with three types of bacteria in both soil types. The bud numbers, fresh weight, and seeds’ dry weight increased in seeds treated with *A. chroococcum* and *B. megaterium* in the sandy soil. Taken together, these findings suggested that the inoculation of plants with PGP bacteria could be used to diminish the implementation of chemical fertilizer and improve the goodness of agricultural products. These findings expand the understanding of the responsive mechanism of microbial inoculation under different soil types, especially at physiological and biochemical levels.

## 1. Introduction

With modern agriculture, especially in the last 50 years, the soil implementation of chemical fertilizers has increased aggressively. The intensive uncontrolled utilization of chemical fertilizers in soil has caused a run of natural issues that includes groundwater pollution, soil quality corruption, and biodiversity reduction. Now, the objective of the agricultural approach is to diminish fertilizer utilization by at least 20% by the year 2030 while guaranteeing the maintenance of soil health and fertility [1]. At the same time, the global world population is predicted to increase in the next years to reach 9 billion in 2050 [2], which places us under a great responsibility to increase crop productivity and meet upcoming food demands. Biofertilizers can be the solution to solve these issues by increasing agricultural productivity, maintaining soil health and vitality, and reducing the accumulative risks of chemical fertilizers on the environment [3].

Nitrogen, phosphorus, and potassium are important macro-elements and are restricted nutrients for the expansion and evolution of plants. A small amount of added nitrogen and phosphorus is consumed by plants due to leaching, denitrification, and volatilization as well as the presence of Fe, Ca, and Al in the soil, which react with P and immobilize it by generating insoluble complexes [4,5]. Changing the unavailability status of these important nutrients could be achieved by several biological processes such as nitrogen fixation, P, K, and silicate solubilization [6,7]. The utilization of microbial biofertilizers is a more eco-friendly strategy as they are ecologically gentler to plants and give plants the benefits of all the soil nutrients [8]. Microorganisms are present near every living thing on the planet since they are ubiquitous in nature. When microorganisms connect with rhizosphere soil as biofertilizers, they colonize it and increase the take-up of plant nutrients [3].

The interaction between plants and rhizosphere bacteria contributes directly to soil fertility and plant health, especially that of plant growth-promoting bacteria (PGPB) as common biofertilizer bio-agents [8,9]. PGPB are free-living strains of bacteria that directly contribute to plant development and root systems via nitrogen fixation, mineral solubilization, and the release of many phytohormones [10]. In recent years, scientists have focused on stabilizing and preparing bacterial consortium types for bettering plant health, growth, yield, and additionally as a biocontrol factor [11]. According to the literature, PGPB is now employed in agriculture, in the cycling of minerals and biofertilizers, as a tool for sustainable agriculture [9]. The prevalent PGPB bacterial genera used in agriculture include *Bacillus*, *Azotobacter*, *Arthrobacter*, *Acinetobacter*, *Beijerinckia*, *Derxia*, *Klebsiella*, *Enterobacter*, *Serratia*, *Gluconacetobacter*, *Alcaligenes*, *Pseudomonas*, *Azospirillum*, *Paenobacillus*, and *Lactobacillus* [10,11,12].

The role that rhizosphere microbes play in the evolution, health, and nourishment of plants makes them of the highest significance. Dong et al. [13] stated that 99.9% of microorganisms in nature exist as microbial consortiums. Also, numerous studies have demonstrated that microbial consortiums composed of multiple species can perform a variety of beneficial tasks that a single microorganism cannot [14]. The direct PGP bacterial mechanism in plants includes solubilizing various useful minerals and releasing them in a valuable form in soil, such as phosphates, potassium, zinc, iron, calcium, and silicon, and also increasing the plant nitrogen uptake through natural fixed nitrogen [15]. They synthesize biological phytohormones like gibberellins, abscisic acid, auxins, ethylene, and cytokinin, and also have a role in siderophore sequestration [16]. The indirect PGP bacterial mechanism in plants involves generating the plant’s systemic resistance and stimulating its immune response through the rhizosphere ecology. The PGP bacteria’s substances such as siderophore, antifungal agents, pigments, mycotoxins, antibiotics, vitamins, organic acids, and volatile compounds activate these protection responses against stress, especially the biotic ones [17,18].

Pea crop (*Pisum sativum* L.) is a valuable leguminous crop and is a rich source of protein [19], phenolics, tannins and flavonoids, and antioxidants [20]. Pea crops are considered a critical crop in sustainable agriculture [21]. Peas are farmed in about 84 countries all over the world. This high predominance of peas is connected to their extraordinary growth, yield, and importance for human nutrition livestock. Pea seeds have massive nutritional aspects, including high protein, carbohydrate, vitamin, phosphorus, and calcium contents [22,23]. Prior studies have mentioned that PGP rhizobacteria improved the growth and yield of various vegetable crops, i.e., potato, tomato, onion, pepper, beans, and lettuce [6,24,25]. Although previous researchers reported that plant growth-promoting bacteria vary in diverse agriculture crops and even in different varieties of the same crops [26], only a few research papers have studied their effects on pea. Chamekh et al. [27] studied the effect of 115 PGP bacterial strains on pea growth under lead toxicity, and demonstrated the enhancing effects of PGP bacteria of plant growth in normal and stress conditions. Shabaan et al. [28] found that treating pea seeds with PGP bacteria improves plant height, shoot and root dry weights, and seed weight under heavy metal stress.

Now, there is an excessive need to explore various ways to enhance the nutrient bioavailability to plants, especially with the increase in our food demands, environmental climatic shifting, and the suffering of the soil from the continuous uncontrolled infiltration of chemical fertilizers. Although biofertilizers could be the most effective eco-friendly solution, there has been limited work on the PGP bacterial impacts on vegetable crop growth and productivity, and more future studies are urgently needed. Therefore, the current study was set up to investigate the impacts of three microbial biofertilizers, *Azotobacter chroococcum*, *Bacillus megaterium*, and *Bacillus cerkularice* PGP bacteria, on pea growth and yield quality in clay and newly reclaimed soil. We also investigated the effects of single and consortium interaction treatments in terms of the phenotype, yield components, and physiological and biochemical responses on pea growth and seed quality in the two soil types. Also, we explained the correlations between the investigated parameters and treatments using cluster analysis.

## 2. Results

### 2.1. Plant Growth-Promoting Properties of Bacteria

Phosphate solubilization, potassium solubilization, IAA production, N-fixation, and ammonia production were all estimated in *Azotobacter chroococcum*, *Bacillus megaterium*, and *Bacillus cerkularice* as plant growth-promoting properties. The isolates have phosphate and potassium solubilization abilities with 9.2 ± 0.6, 16.7 ± 0.5, and 12.5 ± 0.7 mm phosphate solubilization activities and 5.8 ± 0.2, 10 ± 0.8, and 13.7 ± 0.5, potassium solubilization for *A. chroococcum*, *B. megaterium*, and *B. cerkularice*, respectively. The three isolates indicated positive results for both ammonia production and IAA production up to 52.7 ± 1.9, 70.7 ± 1.2, and 50.5 ± 1.8 μg/mL IAA for *A. chroococcum*, *B. megaterium*, and *B. cerkularice*, respectively. Moreover, the nitrogen-fixing activity of *A. chroococcum* was made positive by growing it on a nitrogen-free medium. All three microbes do not have any antagonistic properties on nutrient agar medium.

### 2.2. Microbiological Soil Analysis

The microbiological analysis of the two soil types showed clear differences between the different treatments for the total aerobic bacteria (TAB) and the nitrogen-fixing bacteria (NFB), as elucidated in Figure 1. For both the TAB and NFB, the counts were higher in the clay soil than the sandy soil and the bacterial consortium was higher than the control treatment and the sing treatments. For the total aerobic bacteria, the bacterial counts in the clay and sandy soils were higher in the mixed treatments than the single ones, with the highest number in the bacterial consortium (ABC), giving 143.3 ± 5.4 and 94 ± 4.3 × 10^7^ CFU/g of clay and sandy soil, compared with the non-bacteria-treated soil (CK) at 54 ± 2.9 and 35 ± 3.7 × 10^7^ CFU/g of clay and sandy soil, respectively. For NFB, it was observed that the samples inoculated with *Azotobacter* contained higher NFB counts than the non-inoculated ones. Also, the bacterial consortium ABC recorded the highest count for both clay and sandy soil, giving 86.3 ± 3.8 and 77 ± 1.6 × 10^6^ CFU/g of clay and sandy soil, compared with the non-bacteria-treated soil (CK) at 23.3 ± 2.6 and 15.7 ± 3.09 × 10^6^ CFU/g of clay and sandy soil, respectively.

### 2.3. Phenotypic Criteria and Yield Component Measurements

The morphological responses of pea to inoculation with various types of bacteria in clay and calcareous sandy soil were investigated. The plant length was apparently higher in clay soil than in calcareous sandy soil. In contrast, the root length and number of lateral roots were found to be higher in sandy soil than in clay soil (Table 1). The same trend was observed in the pea seeds treated with the bacterial consortium (ABC) compared to the other treatments and the control regarding the plant length, root length and number or lateral roots, with no significant differences compared with the AB treatment in the root length or number or lateral roots. Pea seeds treated with bacterial consortium (ABC) in the clay soil registered the highest plant length (49 cm) compared to the untreated seeds planted in clay (24.6 cm) or calcareous sandy (27.7 cm) soil. The same trend was observed in the number of lateral roots, with no significant differences to the AB treatment under sandy soil conditions. However, in the sandy soil, the AB treatment showed the longest root length (12.3 cm), with no significant differences compared with *Bacillus cerkularice* (C).

There were significant differences in the shoot fresh and dry weight, and root fresh and dry weight of the pea plants affected by the soil type, various inoculation treatments, and their interaction (Table 2). Sandy soil gave the highest weight of fresh and dry shoots using the bacterial consortium (ABC). Regarding the interaction, the ABC treatment in the clay soil registered the highest value of shoot fresh weight. However, the highest shoot dry weight was observed in sand soil with the AB treatment, with no significant differences with sand soil plus ABC and clay soil plus ABC. In contrast, clay soil gave the highest weight of fresh and dry roots. The bacterial treatment (AC) produced the maximum root fresh weight (1.03 g), while the ABC treatment gave the highest root dry weight (0.28 g), with no significant differences in the AC treatment (0.27 g). The trends in the average root fresh weight were similar to those of the average root dry weight, in which pea cultivated in clay soil and inoculated with bacterial consortium (AC) showed the highest fresh and dry root weight.

The results presented in Table 3 show that the trends in the average bud number, bud fresh weight and seed dry weight were similar to the shoot fresh and dry weights in terms of the impact of soil type, where the obtained values were 26.8, 79.7 g and 8.66 g, respectively. However, clay soil produced the biggest dry weight of buds (3.91 g) compared to the sand soil. The results also showed a highly significant difference (*p* < 0.01) among bacteria inoculation treatments in the bud number, fresh and dry weight, and seed dry weight. The maximum value for the mentioned parameters was obtained when the pea plants were inoculated with the bacterial consortium (AB). The interaction effect showed that planting pea in sandy soil and inoculating it with AB bacteria significantly increased its number of buds, the fresh weight of the buds and the seed dry weight compared to the other treatments. However, the same inoculated treatment, but in clay soil, gave the maximum bud dry weight.

### 2.4. Shoot Physiological and Biochemical Analysis

The physiological and biochemical responses in plant shoots to the effects of different soil types, various bacterial inoculation treatments, and their interactions were determined. The chlorophyll a and b, and carotene contents were greatly increased in clay soil compared to sandy soil, with an increase percentage of 38.42%, 17.09% and 5.33%, respectively. Meanwhile, those of inoculated pea seeds and plants with three different types of bacteria (ABC) were found to be higher than those of the untreated plants, with a reduction percentage of 50%, 72.02% and 75.74%, respectively. Additionally, the interaction results showed that the combination of clay soil and inoculation with ABC bacteria gave the highest values of Chl a and b and carotene, with no significant differences in the sandy-soil-plus-ABC treatment in the Chl b and carotene levels (Figure 2A). In contrast, the interaction between the sand soil and sand-soil-plus-ABC treatment gave the maximum values of glucose, sucrose and starch contents. However, the effects of various bacteria treatments on the Chl a/b and carotene levels shared a similar trend. The percentage increase in sand soil compared to clay soil was 17.44%, 65.44% and 81.91%, respectively, in terms of the mentioned parameters. Regarding to the inoculation treatments, the percentage increase compared to the control treatment was 206.68% for glucose, 298.18% for sucrose, and 213.02% for starch (Figure 2B).

As shown in Figure 3A, differences in the total nitrogen and soluble protein contents between soil types were highly significant. However, no significant difference was found in the free amino acid content. However, it shared the same general trend with Chl a, b, and carotene as well as glucose, sucrose and starch, in terms of the effect of the bacterial inoculation treatments, in which the ABC treatment elicited the biggest value of total nitrogen, free amino acid and soluble proteins (35.25, 46.58 and 46.87 mg/g DW, respectively). Regarding to the impact of the interaction, the clay-soil-plus-ABC treatment produced the maximum value of 40.25 mg/g DW of total nitrogen and 51.87 mg/g DW of soluble proteins, with a percentage increase of 70.33% and 219.79%, respectively, compared to the control plants cultivated in clay soil. However, the ABC treatment with sandy soil registered the highest content of free amino acids, with a percentage increase of 339.47% compared to the untreated plants cultivated in sand soil.

The statistical analysis revealed that there were significantly differences in the phenolic, flavonoid and anthocyanin contents among the soil types, bacteria treatments, and their interactions (Figure 3B and Figure 4A). Sandy soil produced the greatest content of the previous measurements (5.33, 1.54 and 37.64, respectively). Moreover, the maximum flavonoid and anthocyanin contents were obtained when pea plants were treated with the combination of the three types of bacteria (ABC). The percentage reduction in the control plants (CK) was 89.92% in the flavonoids and 73.55% in the anthocyanin. In contrast, the control treatment gave the greatest content of phenolics (7.92). Furthermore, untreated plants cultivated in sand soil showed the largest value of phenolic content (8.92). In contrast, the ABC treatment in clay or sand soil gave the maximum flavonoid contents (3.37 and 3.77, respectively). In addition, peas planted in sand soil and inoculated with the bacterial consortium (ABC) recorded the highest anthocyanin content (66.54).

The total antioxidant, ascorbic acid, nitrate, phosphorus, and potassium responses of pea plants to different soil types, various bacteria treatments, and their interactions were investigated (Figure 4A,B). The total antioxidant content was found to be higher in sandy soil than in clay soil; in contrast, the other parameters were higher in clay soil than in sandy soil. The pea seeds inoculated twice with three types of bacteria (ABC) were superior to the other treatments, with values of 187.54%, 139.38%, 192.17%, 677.14%, and 229.91%, respectively, compared to the untreated plants. In terms of interaction, plants cultivated in sandy soil and treated with ABC bacteria had a significantly greater total antioxidant percentage (112.01%) and ascorbic acid value (49.37 mg/g DW). However, the clay-plus-ABC treatment recorded the greatest nitrate, phosphorus and potassium contents (54.73, 3.29, and 58.75 mg/g DW, respectively). As presented in Figure 5**,** lignin was higher in clay soil than in sandy soil. Additionally, inoculating pea seeds and plants with bacteria reduced their lignin content compared to the control treatment. The same trend was observed for the interaction, where un-inoculated plants (CK) cultivated in clay soil recorded the highest lignin content, followed by sandy soil.

### 2.5. Seed Biochemical Analysis

Mature seeds were used to investigate the contents of soluble sugars, proteins, total carbohydrates, ascorbic acid (vitamine C), and iron to understand the effect of these studied factors on seed quality (Figure 6). Sandy soil produced seeds with significantly greater soluble sugar, total carbohydrate and ascorbic acid contents (11.96, 10.81, and 27.05 mg/100 g DW, respectively). However, clay soil showed higher protein and iron contents (14.39 and 3.59 mg/100 g DW, respectively). Compared to un-inoculated plants, the ABC treatment obtained the highest values of the studied parameters, with percentage increases of 100.36%, 111.25%, 147.97%, 87.57%, and 284%, respectively. The effect of the interaction between the soil type and various types of bacteria was highly significant. Inoculating plants with three types of bacteria (ABC) in sandy soil increased their contents of soluble sugars and ascorbic acid (15.26 and 36.29 mg/100 g DW, respectively). However, the clay-soil-plus-ABC treatment increased their protein, total carbohydrate, and iron contents (19.24, 15.38, and 6.02 mg/100 g DW, respectively).

### 2.6. Cluster Analysis

Cluster analysis, also known as multivariate classification, enhances the interpretation process by indicating the degree of similarity between more than two treatments (Figure 7) or criteria (Figure 8). Consequently, it was applied to the acquired data in order to identify all potential positive and negative correlations (similarities) between the plant responses under the effect of treatments and plant/seed morpho-physiological criteria.

Significant to highly significant positive correlations were found among the applied bacterial treatments in the case of both the used clay soil and calcareous sandy soil (Figure 7A,B). In the clay soil, plants responded to the bacterial treatments similarly (r^2^ = 0.90) when inoculated with the control, A, B, and C (group A) treatments, while the second inoculation group (B) (r^2^ = 0.88) showed significant similarities in plants treated using the BC, ABC, AC, and AB treatments (Figure 7A). On the other hand, plants grown on calcareous sandy soil responded to the bacterial treatments similarly (r^2^ = 0.955) in group A when inoculated with the control, BC, B, and C treatments. A less divergent response (r^2^ = 0.935) of the plants to the bacterial treatments was exhibited among group (B), including plants inoculated with the ABC, AC, A, and AB treatments (Figure 7B).

Two major groups of criteria were identified for plants grown either in clay or calcareous sandy soil (Figure 8A,B). There were significant positive correlations between group (A) components, including all seed and plant morphological and physio-biochemical criteria, excluding lignin and phenolics (group C) which were negatively correlated (r^2^ = −0.8) with group (A). It is worthy to mention that regarding calcareous sandy soil, group (A), including plant criteria affected by bacterial treatments, was subdivided into subgroup (B), including phosphorus and potassium (r^2^ = 0.2).

## 3. Discussion

These days, we face an increasing requirement for crop production around the world to meet food demands and food industrial processes, which is expected to reach almost 9 billion humans by 2050 [29]. Invigorated by the expanding agriculture demand, and mindful of the negative impacts of excessive chemical utilization in current horticulture practices, world-wide farming is moving towards more economical and eco-friendlier approaches [8,30]. Fertilization is considered a fundamental route to enhance the nutrient accessibility of soil to plants. Although utilizing fertilizers is dominant as one of the foremost important components in improving plant yields in modern agriculture, the overwhelming employment of chemical fertilizers have caused an assortment of natural and biological issues [25]. Therefore, the rational use of chemical fertilizers or more intensive use of natural fertilizers would be more beneficial [31]. Various types of soil microorganisms play an important role in enhancing plant and soil health. Several researchers have reported that a symbiotic relationship established between beneficial soil microbes and plants can help the plants in nitrogen earning, water absorbance, and enhancing tolerance to numerous stress [32,33,34].

During our research, we investigated the impacts of three microbial biofertilizers, *Azotobacter chroococcum*, *Bacillus megaterium*, and *Bacillus cerkularice* PGP bacteria, on pea growth and seed quality in clay and newly reclaimed soil, with single and consortium treatments. Our results showed that the three bacterial types have promising efficiency in enhancing the pea plants’ growth and production; however, the bacterial consortium treatments were more effective treatments than single bacteria. Also, we observed that the plant length was higher in clay soil than in calcareous sandy soil. In contrast, the root length and number of lateral roots were found to be higher in sandy soil than in clay soil. The bacterial consortiums (ABC) and (AB) were the best bacterial treatments in terms of the phenotype, yield components, and physiological and biochemical responses.

The abundance of nitrogen within the climate is roughly 78%, and plants cannot acclimatize it. Numerous PGPRs have been distinguished that can carry out atmospheric nitrogen fixation, especially when associated with legumes, and are free in soil. Nitrogen-fixing microorganisms improve soil quality and increase seed germination, seedling quality, root length, and plant development. A few cases of advantageous nitrogen fixers are *Rhizobium*, *Azotobacter*, *Azorhizobium*, *Pseudomonas* and *Bradyrhizobium* [35,36]. PGP bacteria also have unique properties in phosphate and potassium solubilization, and can dissolve insoluble phosphorus and potassium in the soil by up to 90% in some cases [37,38]. The PGP bacteria that produce IAA are known as phytostimulators, and the generation of auxins is one of their most regularly detailed functions for the advancement of plant development. It has been found that 80% of microorganisms associated with the rhizosphere can create this metabolite, particularly the bacterial genus *Bacillus* [38]. Auxin controls meristematic tissue differentiation, root elongation, seed differentiation, pathogen infection and fruit development. When PGP bacteria were inoculated to the plant, they raised the number of root hairs and lateral roots, which resulted from bacterial auxin generation. It is well known that lignification generates dropping of the cell wall extensibility, which restricts cell enlargement [39]. In the current study, in general, PGP bacterial application has a vital role in enhancing plant elongation and growth, caused by controlling lignification, in terms of an adequate lignin content compared with un-inoculated plants.

Here, we found that a nitrogen-fixing bacterial strain (*Azotobacter chroococcum*) plus the highest phosphate-solubilizing bacterial strain (*Bacillus megaterium)* gave the highest bud number, fresh and dry weight, and seed dry weight results. Mixed strain formulations recorded without doubt way better comes than individual strains because they can advance plant development and improvement and offer assistance in combating different stresses and diseases [32]. These strains in the rhizosphere help plants in their growth through enzymatic and other metabolic mechanisms [40]. Previous research has reported similar results, where inoculation enhanced plant height, leaf number, shoot fresh and dry weight, and root length compared to non-inoculation in cucumber, lettuce, onion, tomato, potato, bean, *Ficus benjamina*, and rice [20]. Moreover, in the current study, the yield-attributed traits of pea, including the number of buds per plant, fresh and dry weight of buds, and seed dry weight were increased significantly, affected by the synergistic inoculation of seed pea. These results were in accordance with Bizos et al. [41], who mentioned that bacterial inoculation enhanced yield by increasing the number of flowers and fruits under calcareous soil conditions. Potato yield was increased significantly in inoculated plants with tree types of bacteria compared to non-inoculated [6]. It was reported that a consortium of various types of bacteria recorded better results compared to mono and dual inoculations in terms of plant parameters and grain yield [42]. In the same line, using bioorganic fertilizer for a long period in an apple orchard increased the apple yield, modified the structure of the microbial community in the soil and improve soil characteristics [31,43].

Our results corroborated previous findings that PGPR-inoculated plants significantly increased levels of chlorophyll a and b, and the total carotenoid concentration compared to un-inoculated plants [2]. This might be due to the improved availability of N and P, which had a positive effect on the photosynthetic rates in plant leaves [44]. Previous investigation has confirmed that the major mechanisms for rhizosphere microbes include alterations in root architecture, osmolyte production, antioxidants, activities of phytohormones, extracellular polymeric substances (EPS), and volatile organic compounds [32,40]. Results of this study showed that the total antioxidant value was found to be higher in inoculated pea plants. Similar results were reported by EJaz et al. [2]. These findings agree with a previous study, which reported that inoculating maize plants with arbuscular mycorhizal fungi (AMF) improved their antioxidant enzyme activity compared to non-inoculated plants [45]. In addition, previous studies have demonstrated that calcareous soils are usually nutrient-deficient [46,47]. Moreover, Tahir and Marschner [48] reported that clay soils are enriched in nutrients more than sandy soils; however, the bioavailability of nutrients for plants absorbance is less. In addition, the obtained results showed that total nitrogen, P, K, proteins, free amino acid and soluble proteins increased after inoculation with the three types of bacteria. A high nutrients content with inoculation with ABC bacteria improved the pea plant growth and seed yield and this might be related to a high nutrient uptake by the roots of pea plant due to the increased root length and number or lateral roots [42]. Zahran et al. [6] showed that potato tuber proteins and starch increased in plants inoculated with various types of bacteria in the calcareous soil compared to CK plants. This might be as a result of a high content of N, P and K in leaves, which donate efficiently to the protein and starch composition [49]. Moreover, N is directly included in amino acid synthesis, which are the essential compounds for protein synthesis [50]. These results were supported by Saboor et al. [46], who concluded that AM-inoculation could increase the nutrient uptake in calcareous soil by improving the root surface area. Moreover, the obtained results in the calcareous soil reveled that plant roots, nutrient contents, proteins and total antioxidants were higher in the inoculated plants compare to non-inoculated ones. This might be due to an increase in the number of beneficial soil microbes in the root rhizosphere. Samuel et al. [51] reported that soils containing lime cause significant enhancements in its chemical and biological reactions and in its microbiological processes. This results in changes in many chemical compounds’ solubility, an improvement in plant roots, environmental development, an increase in soil microbial biomass, including microbial dynamics and diversity, and a significant improvement in enzyme activities.

Furthermore, response indicators have been grouped based on their similarity using the cluster analysis technique [52]. The eight treatments in the current study were categorized into two clusters for both types of soil. For the pea plant responses under control and single treatments, the similarity distance showed a close resemblance (nonsignificant differences), while consortium treatments display similar effects to each other, indicating their significance in improving plant quality criteria [14]. Ultimately, a straightforward multivariate correlation analysis was used to shed light on all of these investigated plant and seed morphological and physio-biochemical criteria, and the results provide compelling evidence regarding the significance of applying bacterial treatments to pea plant in order to augment its quality and yield when grown in such poor and reclaimed soil. In the current trail, the thirty-four assessed criteria were divided into two clusters. Phenolics and lignin were included in one cluster. This indicated that validation in approving bacterial treatments’ effectiveness in enhancing pea plant quality in the present study was correlated to reduce aging factors, e.g., phenolic and lignin content [39].

## 4. Materials and Methods

### 4.1. Microbial Consortium

Three strains of plant growth-promoting rhizospheric bacteria; *Azotobacter chroococcum* 14346, *Bacillus megaterium* 670, and *Bacillus cerkularice* 692 were utilized during this experiment as plant growth-promoting bacteria that were kindly obtained from the Agriculture Research Center, Egypt [6]. *Bacillus megaterium* and *B. cerkularice* were kept on a nutrient agar (NA) medium while *A. chroococcum* was maintained on nitrogen-free medium aerobically, preserved at 4 °C ± 1 °C, and sub-cultured every 4 weeks [53]. *Azotobacter chroococcum*, *B. megaterium*, and *B. cerkularice* were examined for antagonistic interactions on NA plates prior to use (no antagonistic activities were detected). Prior to the experiments, the microbial strains were activated on the same medium for one day at 28 °C ± 1 °C in 200 rpm rotary incubator till 0.1 OD660 (Lab-Line 3597 Orbital Environmental Shaker, Thermo Fisher Scientific, San Francisco, CA, USA). Following that, the biomass was gathered, centrifuged at 6000× *g* for 10 min (IEC CRU-5000 Refrigerated Floor Centrifuge, Thermo Fisher Scientific, Middletown, VA, USA), and suspended in sterile saline with 1 × 10^6^ CFU/mL.

### 4.2. Plant Growth-Promoting Properties of Bacteria

The phosphate-solubilizing activities of PGP bacteria were tested according to Pikovskaya [54] and Khanghahi et al. [55]. Bacterial strains were inoculated on the surface of Pikovskaya’s solid agar medium (glucose, 1%; Ca_3_(PO_4_)_2_, 0.5%; MgCl_2_·6H_2_O, 0.5%; MgSO_4_·7H_2_O, 0.025%; (NH_4_)_2_SO_4_, 0.01%; KCl, 0.02%; agar, 1.5% and 100 mL distilled water), then cultures were incubated for 48 h at 30 °C ± 1 in a static incubator (B5050, Heraeus, Gemini BV, Prinses Beatrixlaan 301, 7312 DG Apeldoorn, The Netherlands). Phosphate-solubilizing activities were detected as halo zones surrounding the microbial growth. The potassium-solubilizing activities of PGP bacteria were detected following the method of Shanware et al. [56]. Bacterial strains were loaded onto the surface of Aleksandrov solid agar medium (glucose, o.5%; mica, 0.1%; Ca_3_(PO_4_)_2_, 0.01%; FeCl_3_, 0.001%; MgSO_4_·7H_2_O, 0.025%; CaCO_3_, 0.01%; agar, 1.5%; and 100 mL distilled water), then cultures were incubated for 48 h at 30 °C ± 1 in a static incubator. The potassium-solubilizing activities were detected as halo zones surrounded the microbial growth. For indole acetic acid (IAA) revelation in PGP bacteria, the isolates were grown in a nutrient broth medium supplemented with 0.02% L-tryptophan at 30 °C ± 1 and 150 rpm for 48 h. After incubation, culture filtrate was centrifuged at 6000× *g* for 10 min, the supernatant was gathered and treated with Salkowski reagent (1:1, *v*:*v*) and left for 20 min at 30 °C with stirring. The developed pink color was subjected to a T60 UV spectrophotometer (535 nm) over the 100–1100 nm range, with 2 nm fixed silt (PG Instruments Lmited, Woodway lane, Alma park, Leicestershire LE17 5BH, United kingdom), and the IAA values were calculated according to the standard curve (Sigma-Aldrich, St. Louis, MO, USA) [8,57]. Also, the nitrogen-fixing capability of *Azotobacter chroococcum* was confirmed on a nitrogen-free medium following the method of Doroshenko et al. [58]. For ammonia production by PGP bacteria, bacterial isolates were inoculated in 1% peptone water, and incubated for 48 h at 30 °C. Before incubation, culture filtrate was gathered and centrifuged at 6000× *g* for 10 min, and the supernatant was treated with Nessler’s reagent (1:1, *v*:*v*). The brown-yellow color reflects a positive result for ammonia generation [59].

### 4.3. Plant Material and Soil Preparations

Seeds of pea (*Pisum sativum* L.), Master B cultivar was used in the current investigation at Assiut Agriculture Research, Assiut governorate, Egypt (27°10′60.00″ N, 31°09′60.00″ E). Two types of soils were collected (Table 4); the first type was from the old agricultural lands in the Assiut government, Egypt. The second one was from newly reclaimed lands at Assiut Agricultural Research Station, Assiut, Egypt. Composite soil samples (0–15 cm depth) of two types were collected before sowing and analyzed in the Agriculture Research Center. Soil samples were transferred in the oven (Heraeus VT 5042 EK Vacuum Oven-Gemini BV, Prinses Beatrixlaan 301, 7312 DG Apeldoorn, The Netherlands) at 40 °C for 48 h for complete drying and passed via a 1 mm sieve. The soil texture was classified as Typic Torrifluvents according to USDA Soil Taxonomy [60]. Field capacity was determined by using the pressure plate apparatus [61]. The soil pH and electrical conductivity were estimated using a pH-ORP-Conductivity-TDS-TEMP Bench Meter (AD8000, Szeged, Hungary); the saturation capacity, total calcium, nitrogen, organic matter, soluble cations and anions, phosphorus, and potassium were estimated following the protocol of Jackson [62].

### 4.4. Experimental Design

The existing experiment was performed in a greenhouse as a randomized complete design in a split-plot, with three replications, considering soil type (clay soil and sandy calcareous soil) as the main plot (Figure 9). However, the second factor was assigned in a sub-plot to bacteria treatments. The used bacteria were *Azotobacter chroococcum* (A), *Bacillus megaterium* (B), and *Bacillus cerkularice* (C), in a concentration 9 × 10^8^ CFU/mL. Five healthy pea seeds were sown in a plastic pot containing 10 kg of used soil. The inoculation with the three different types of bacteria was performed twice, at sowing of the seeds and 15 days after sowing the seeds. Plants were irrigated to field capacity.

### 4.5. Microbiological Soil Analysis

For the quantitative microbiological analysis of the rhizosphere soil, total aerobic bacteria (TAB), and nitrogen-fixing bacteria (NFB) were detected. The pea plants were uprooted and gently shaken to remove superfluous soil, then ten grams of root-adhering soil were transferred into a flask containing 90 mL sterilized saline solution, and vortexed (ZX, VELP Scientifica, New York, NY, USA) for 5 min. Then, rhizosphere stock was diluted until the suitable concentration. For total aerobic bacteria, 1 mL of the rhizosphere soil solution was passed into a sterilized Petri dish and mixed with nutrient agar medium, incubated at 30 ± 1 °C for 48 h (Heraeus B5050E Incubator, Gemini BV, Prinses Beatrixlaan 301, 7312 DG Apeldoorn, The Netherlands) [63]. For nitrogen-fixing bacteria, 1 mL of the rhizosphere soil solution was passed into a sterilized Petri dish and mixed with nitrogen-free mannitol medium, and incubated at 30 ± 1 °C for 48 h (Heraeus B5050E Incubator, Gemini BV, Prinses Beatrixlaan 301, 7312 DG Apeldoorn, The Netherlands) [58]. Total counts of TAB and NFB bacteria were expressed as CFU/g soil.

### 4.6. Phenotypic Criteria and Yield Components Measurements

Ten pea plants in each biological replicate were randomly selected and harvested after 90 days from sowing seeds. The traits of the plant length (cm), root length (cm), number of lateral roots, shoot fresh weight and dry weight (g), and root fresh weight and dry weight (g) were measured. Also, the yield components traits including the number of buds, the bud fresh weight and dry weight (g), and seed dry weight (g) were estimated.

### 4.7. Physiological and Biochemical Analysis

The materials used in the physiological standard curves were puncher from Merck (Darmstadt, Germany) and Sigma-Aldrich (USA). Chlorophyll a/b, and carotenoids were detected by transferring dry leaves in 5 mL 95% ethyl alcohol and incubating them in a 60–70 °C water bath. The absorbances were taken at three wavelengths, 663, 644, and 452 nm, using a T60 UV spectrophotometer over the 100–1100 nm range, with 2 nm fixed silt (PG instruments, Jena, Germany), and the quantities were estimated using Lichtenthaler [64] equations. Carbon metabolism was evaluated in the ethanolic extract through the detection of glucose and fructose following Halhoul and Kleinberg [65], and sucrose (mg/g DW) following Van Handel [66], using anthrone–sulfuric acid reagent, 0.2 g in 30 mL dist. water, 8 mL ethanol 99%, and 100 mL H_2_SO_4,_ and estimated at 625 nm. However, perchloric extracts were utilized in starch estimation (mg/g DW) using anthrone–sulfuric acid reagent as recommended by Schlegel [67]. Nitrogen metabolism was also assessed by some criteria, viz., total nitrogen by the protocol established by Lang [68], accumulated amino acids using ninhydrin reagent in a weight of 0.25 g in 100 mL methanol 99%, and estimated at 580 nm following the protocol of Lee and Takahashi [69]. Then, proteins were estimated using diluted Folin–Ciocalteu reagent (1:2 *v*/*v*) and estimated at 750 nm following the protocol of Lowry et al. [70], respectively. Metabolic molecules like phenolics (Folin–Ciocalteu reagent, and measured at 725 nm), flavonoids (aluminum chloride reagent, and measured at 415 nm), and anthocyanin (Methanol-HCl, and measured at 530 and 657 nm), were estimated following the methods of Kofalvi and Nassuth [71], Zou et al. [72], and Krizek et al. [73], respectively. Non-enzymatic metabolites like ascorbic acid (ASA) as antioxidants were detected using 5% trichloroacetic acid and detected at 760 nm following Ellman’s [74] protocols. The phosphorus content was quantified spectrophotometrically via the protocols of Fogg and Wilkinson [75] using ammonium molybdate–sulfuric acid reagent and detected at 660 nm. Also, the nitrate content was estimated using the method of Cataldo et al. [76]. Potassium was quantified using the flame emission technique (Carl-Zeiss DR LANGE M7D flame photometer, Carl-Zeiss AG, Jena, Germany) as recommended by Williams and Twine [77].

### 4.8. Statistical Analysis

The statistical analysis for the collected data was performed by using Statistix8.1 software [78]. Means separation was performed using Tukey’s test at a 5% probability level and data are presented as the mean ± standard deviations. Multivariate classification (cluster analysis) was performed by using PAST software v.2.11 [79] to improve the interpretation process for revealing the degree of similarity among more than two treatments or criteria.

## 5. Conclusions

The effects of plant growth-promoting bacteria on pea plants in different soil types, with single and consortium inoculations, were extremely significant. Phosphate solubilization, potassium solubilization, IAA production, N-fixation, and ammonia production were the common characteristic traits of the three PGP bacteria *Azotobacter chroococcum*, *Bacillus megaterium*, and *Bacillus cerkularice.* The three types were highly promising PGP biofertilizers, with a high established efficiency around the plant roots demonstrated by the microbiological soil analysis. The three bacterial consortium and AB consortium treatments were the most effective treatments and enhanced the phenotypic properties of pea including the plant length, root length, number of lateral roots, shoot fresh weight and dry weight, root fresh weight and dry weight, and the yield component traits (number of buds, bud fresh weight and dry weight, and seed dry weight). The results demonstrated that the bacterial consortium of the three bacterial types increased the physiological and biochemical constituents in pea shoot and seeds in terms of chlorophyll, carotenoids, carbon metabolism (glucose, fructose, sucrose, and starch), nitrogen metabolism (total nitrogen, amino acids, and proteins), metabolic molecules (phenolics, flavonoids, and anthocyanin), ascorbic acid, phosphorus, potassium, and nitrate contents. These findings indicate the highly beneficial effects of microbial biofertilizer consortiums on enhancing plant growth and productivity. However, the possible utilization of these consortiums in other vegetables as a tool of sustainable agriculture needs more future research.

## Figures and Tables

**Figure 1 plants-12-03931-f001:**
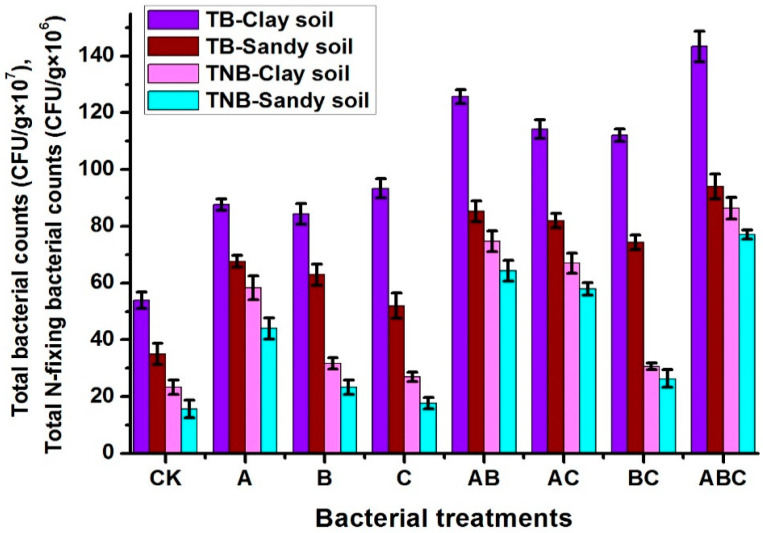
Total aerobic bacterial counts (TB × 10^7^ CFU/g); total nitrogen-fixing bacterial counts (TNB × 10^6^ CFU/g) for the control treatment CK (no bacterial addition); and single treatments A (*A. chroococcum)*, B (*B. megaterium*), and C (*B. cerkularice*); and consortium treatments AB (*A. chroococcum* and *B. megaterium*), AC (*A. chroococcum* and *B. cerkularice*), BC (*B. megaterium* and *B. cerkularice*), and ABC (*A. chroococcum* and *B. megaterium* and *B. cerkularice*).

**Figure 2 plants-12-03931-f002:**
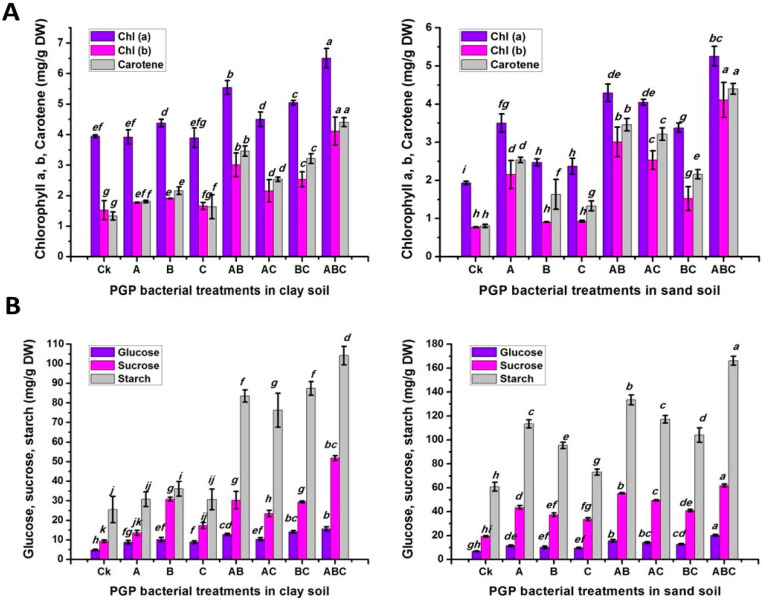
Chlorophyll a/b and carotene (**A**); glucose, sucrose and starch (**B**); and responses of pea inoculated with various types of bacteria in clay and calcareous sandy soil. Control treatment CK (no bacterial addition); single treatments A (*A. chroococcum)*, B (*B. megaterium*), and C (*B. cerkularice*); and consortium AB (*A. chroococcum* and *B. megaterium*), AC (*A. chroococcum* and *B. cerkularice*), BC (*B. megaterium* and *B. cerkularice*), and ABC (*A. chroococcum* and *B. megaterium* and *B. cerkularice*). Diverse lowercase letters, a, b, c, etc., represent significant variances (*p* ≤ 0.05).

**Figure 3 plants-12-03931-f003:**
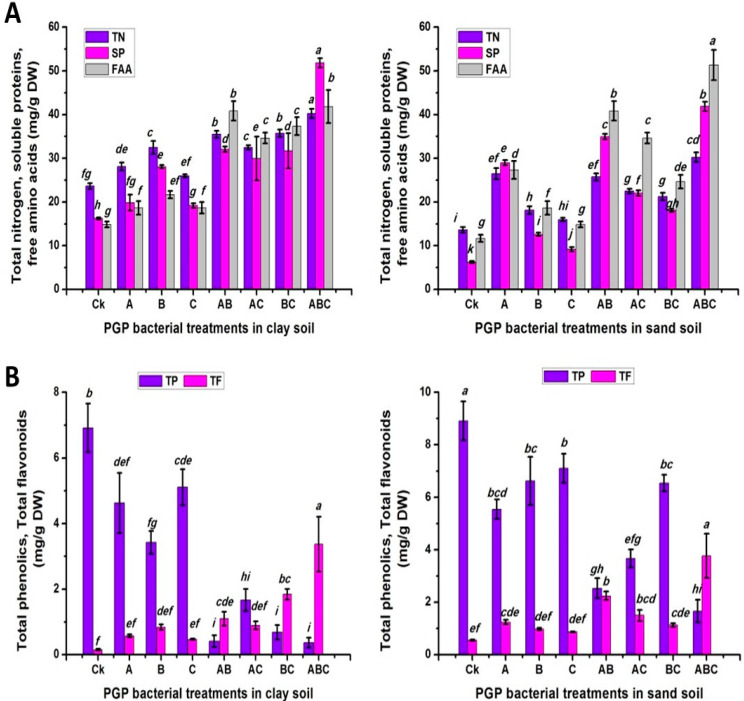
Total nitrogen, soluble proteins, and free amino acids (**A**); phenolic and flavonoid (**B**) responses of pea inoculated with various types of bacteria in clay and calcareous sandy soil. Control treatment CK (no bacterial addition); single treatments A (*A. chroococcum)*, B (*B. megaterium*), and C (*B. cerkularice*); and consortium treatments AB (*A. chroococcum* and *B. megaterium*), AC (*A. chroococcum* and *B. cerkularice*), BC (*B. megaterium* and *B. cerkularice*), and ABC (*A. chroococcum* and *B. megaterium* and *B. cerkularice*). Diverse lowercase letters, a, b, c, etc., represent significant variances (*p* ≤ 0.05).

**Figure 4 plants-12-03931-f004:**
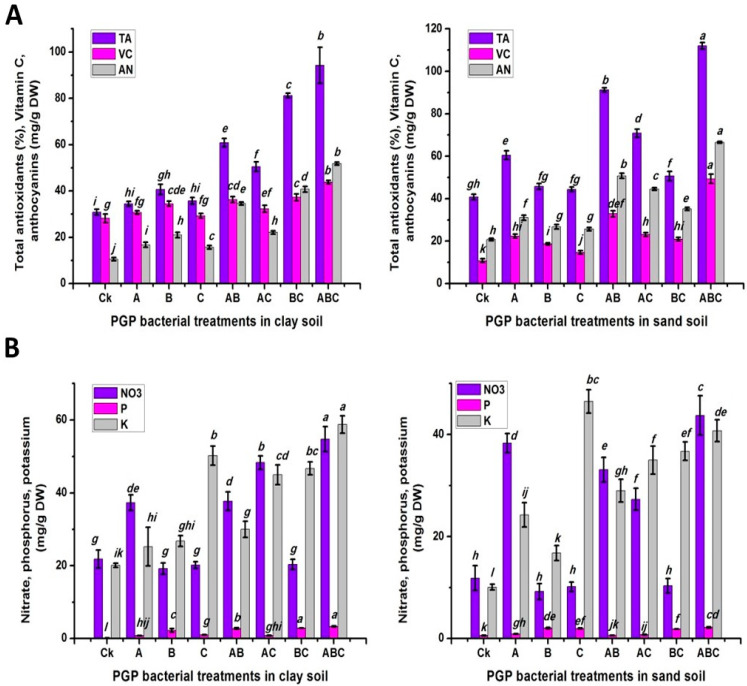
Total antioxidant, vitamin C, and anthocyanin (**A**); nitrate, phosphorus, and potassium (**B**) responses of pea inoculated with various types of bacteria in clay and calcareous sandy soil. Control treatment CK (no bacterial addition); single treatments A (*A. chroococcum)*, B (*B. megaterium*), and C (*B. cerkularice*); and consortium treatments AB (*A. chroococcum* and *B. megaterium*), AC (*A. chroococcum* and *B. cerkularice*), BC (*B. megaterium* and *B. cerkularice*), and ABC (*A. chroococcum* and *B. megaterium* and *B. cerkularice*). Diverse lowercase letters, a, b, c, etc., represent significant variances (*p* ≤ 0.05).

**Figure 5 plants-12-03931-f005:**
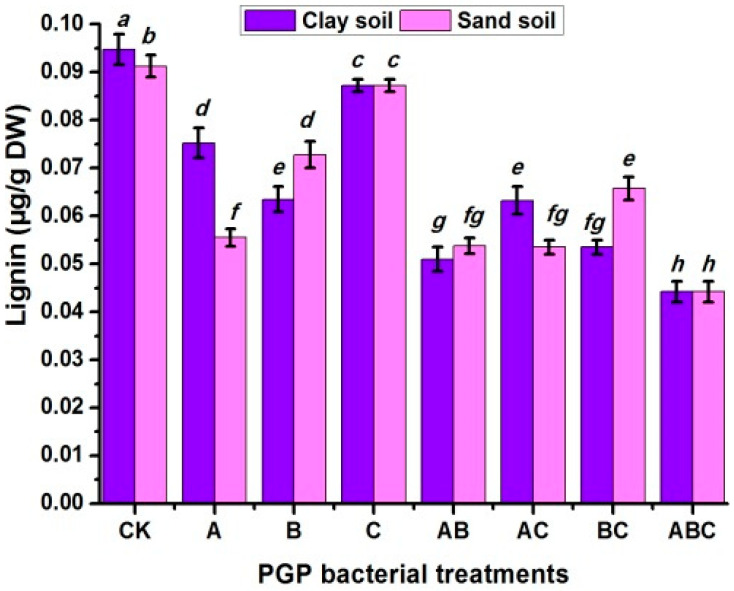
Lignin responses of pea inoculated with various types of bacteria in clay and calcareous sandy soil. Control treatment CK (no bacterial addition); single treatments A (*A. chroococcum)*, B (*B. megaterium*), and C (*B. cerkularice*); and consortium treatments AB (*A. chroococcum* and *B. megaterium*), AC (*A. chroococcum* and *B. cerkularice*), BC (*B. megaterium* and *B. cerkularice*), and ABC (*A. chroococcum* and *B. megaterium* and *B. cerkularice*). Diverse lowercase letters, a, b, c, etc., represent significant variances (*p* ≤ 0.05).

**Figure 6 plants-12-03931-f006:**
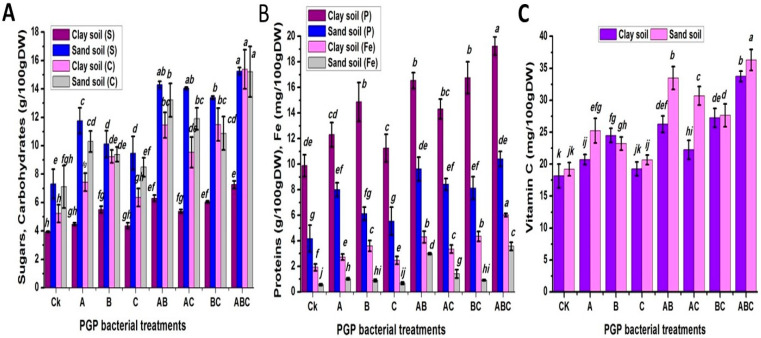
Seed analysis of effect of compounds, including sugars, carbohydrates (**A**), proteins, iron (**B**), and ascorbic acid (**C**), on responses of pea inoculated with various types of bacteria in clay and calcareous sandy soil. Control treatment CK (no bacterial addition); single treatments A (*A. chroococcum)*, B (*B. megaterium*), and C (*B. cerkularice*); and consortium treatments AB (*A. chroococcum* and *B. megaterium*), AC (*A. chroococcum* and *B. cerkularice*), BC (*B. megaterium* and *B. cerkularice*), and ABC (*A. chroococcum* and *B. megaterium* and *B. cerkularice*). Diverse lowercase letters, a, b, c, etc., represent significant variances (*p* ≤ 0.05).

**Figure 7 plants-12-03931-f007:**
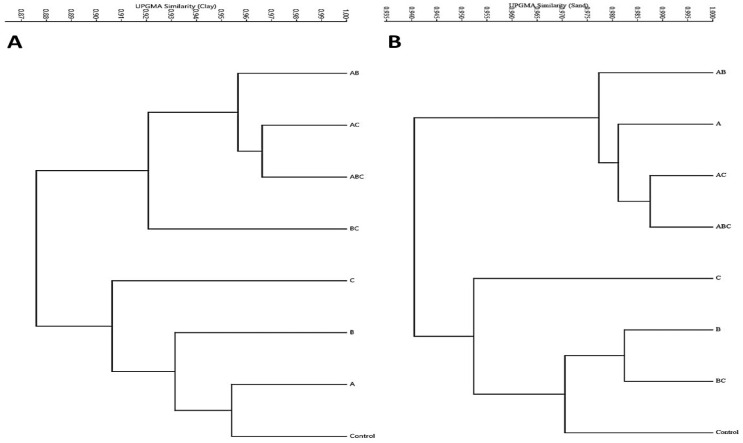
Multivariate cluster analysis of the plant–microbe interaction based on the plants’ phenotype, yield components, physiological and biochemical responses when grown on two different soils: clay (**A**) and calcareous sandy soil (**B**). Abbreviations; control treatment (no bacterial addition), single treatments A (*A. chroococcum*), B (*B. megaterium*), and C (*B. cerkularice*); and consortium treatments AB (*A. chroococcum* and *B. megaterium*), AC (*A. chroococcum* and *B. cerkularice*), BC (*B. megaterium* and *B. cerkularice*), and ABC (*A. chroococcum* and *B. megaterium* and *B. cerkularice*).

**Figure 8 plants-12-03931-f008:**
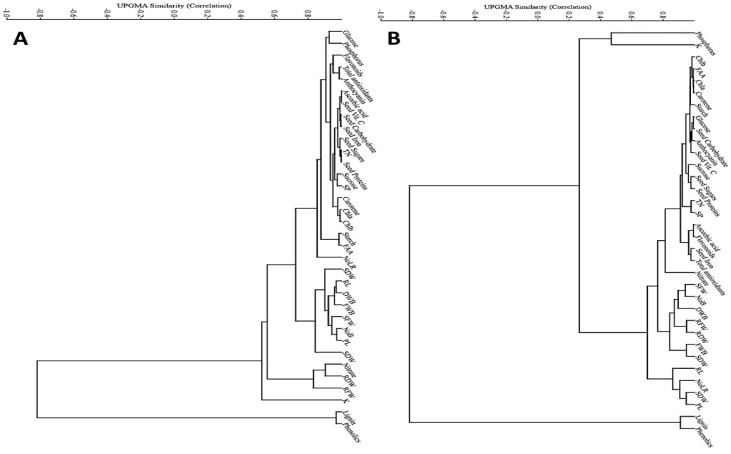
Multi-variate cluster analysis of plant and seed morphological and physio-biochemical criteria based on microbes’ effects when plants are grown on two different soils: clay (**A**) and calcareous sandy soil (**B**). Abbreviations; Vit. C: vitamin C; TN: total nitrogen; SP: soluble protein; Chla: chlorophyll a; Chlb: chlorophyll b; NoLR: number of lateral roots; SDW: shoot dry weight; RL: root length; DWB: dry weight of buds; FWB: fresh weight of buds; SFW: shoot fresh weight; NoB: number of buds; PL: plant length; SDW: seed dry weight; RDW: root dry weight; RFW: root fresh weight; K: potassium.

**Figure 9 plants-12-03931-f009:**
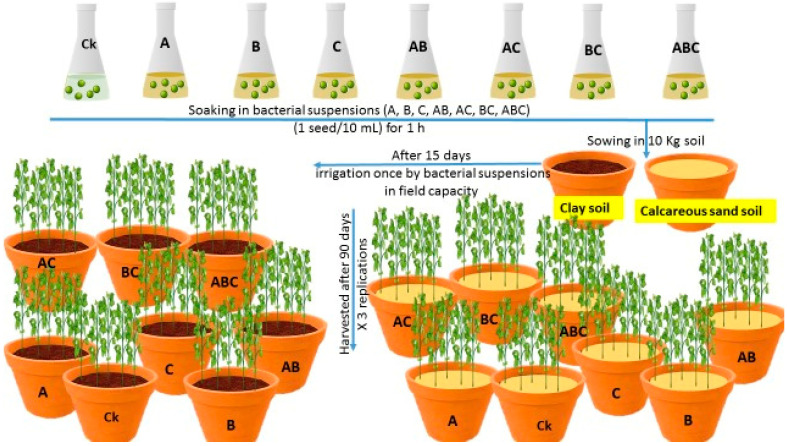
Summarization of the harvest intervals and data collected throughout the study from transplanting to finalizing. Control treatment CK (no bacterial addition); single treatments A (*A. chroococcum*), B (*B. megaterium*), C (*B. cerkularice*); and consortium treatments AB (*A. chroococcum* and *B. megaterium*), AC (*A. chroococcum* and *B. cerkularice*), BC (*B. megaterium* and *B. cerkularice*), and ABC (*A. chroococcum* and *B. megaterium* and *B. cerkularice*).

**Table 1 plants-12-03931-t001:** Plant length (cm), root length (cm), and no. of lateral roots of pea (*Pisum sativum* L.) under the effect of single and consortium plant growth-promoting bacteria (*Azotobacter chroococcum*, *Bacillus megaterium* and *Bacillus cerkularice*) in clay and new reclaimed sandy soil.

**Scheme.**	**Ck**	**A**	**B**	**C**	**AB**	**AC**	**BC**	**ABC**	**Mean**
	**Plant Length (cm)**								
Clay	24.6 ± 0.6 ^h^	38.4 ± 0.9 ^c^	31.4 ± 1.0 ^ef^	30.2 ± 0.8 ^fg^	44.5 ± 0.6 ^b^	36.7 ± 1.1 ^c^	33.7 ± 0.6 ^de^	49.0 ± 1.0 ^a^	36.1 ^a^
Sand	27.7 ± 0.6 ^gh^	33.2 ± 0.7 ^def^	35.2 ± 0.6 ^cd^	36.1 ± 0.7 ^cd^	37.7 ± 1.4 ^c^	35.0 ± 1.3 ^cd^	35.40 ± 1.3 ^cd^	38.2 ± 1.4 ^c^	34.8 ^b^
Mean	26.2 ^e^	35.8 ^c^	33.3 ^d^	33.1 ^d^	41.1 ^b^	35.9 ^c^	34.5 ^cd^	43.6 ^a^	
**Root length (cm)**									
Clay	6.5 ± 0.3 ^j^	9.3 ± 0.1 ^efgh^	8.2 ± 0.3 ^ghi^	8.0 ± 0.5 ^hij^	10.7 ± 0.7 ^bcde^	9.0 ± 0.5 ^fgh^	8.4 ± 0.3 ^ghi^	10.4 ± 0.5 ^cdef^	8.8 ^b^
Sand	7.0 ± 0.2 ^ij^	11.0 ± 0.1 ^abcd^	11.0 ± 0.8 ^abcd^	11.9 ± 0.2 ^ab^	12.3 ± 1.0 ^a^	10.5 ± 0.5 ^bcde^	9.6 ± 0.5 ^defg^	11.8 ± 0.4 ^abc^	10.6 ^a^
Mean	6.8 ^e^	10.2 ^bc^	9.6 ^cd^	9.9 ^c^	11.5 ^a^	9.7 ^cd^	9.0 ^d^	11.1 ^ab^	
**No. of lateral roots**									
Clay	11.0 ± 1.0 ^ij^	12.3 ± 0.6 ^hi^	15.6 ± 0.7 ^defg^	14.0 ± 1.0 ^gh^	17.1 ± 1.0 ^cdef^	17.6 ± 0.5 ^cde^	14.7 ± 0.6 ^fgh^	22.2 ± 1.4 ^a^	15.6 ^b^
Sand	9.2 ± 0.8 ^j^	17.7 ± 0.6 ^cd^	14.9 ± 0.9 ^efgh^	16.2 ± 0.7 ^defg^	21.1 ± 0.9 ^ab^	15.3 ± 0.8 ^defg^	17.7 ± 1.5 ^cd^	19.0 ± 1.1 ^bc^	16.4 ^a^
Mean	10.1 ^c^	15.0 ^b^	15.3 ^b^	15.1 ^b^	19.1 ^a^	16.5 ^b^	16.2 ^b^	20.6 ^a^	

CK: control; A: *A. chroococcum*; B: *B. megaterium*; C: *B. cerkularice*. AB, AC, BC, and ABC are microbial consortiums of the PGP bacteria. Different superscript letters indicate statistically significant differences among soil type, various bacteria and their interaction. Each value represents an average value of three replicates ± SD and averages were compared using LSD at *p* ≤ 0.05.

**Table 2 plants-12-03931-t002:** Shoot fresh weight (g), shoot dry weight (g), root fresh weight (g) and root dry weight (g) of pea (*Pisum sativum* L.) under the effect of single and consortium plant growth-promoting bacteria (*Azotobacter chroococcum*, *Bacillus megaterium* and *Bacillus cerkularice*) in clay and new reclaimed sandy soil.

**Soil**	**Ck**	**A**	**B**	**C**	**AB**	**AC**	**BC**	**ABC**	**Mean**
	**Shoot Fresh Weight (g)**								
Clay	12.9 ± 0.8 ^i^	28.6 ± 1.7 ^fg^	30.8 ± 1.0 ^def^	25.0 ± 1.4 ^gh^	43.4 ± 1.5 ^b^	32.7 ± 1.5 ^cde^	22.2 ± 0.7 ^h^	51.0 ± 1.2 ^a^	30.9 ^b^
Sand	23.7 ± 0.8 ^h^	29.8 ± 0.8 ^ef^	35.4 ± 1.2 ^c^	25.4 ± 1.4 ^gh^	41.7 ± 1.3 ^b^	34.7 ± 1.3 ^cd^	30.3 ± 1.2 ^ef^	42.5 ± 1.3 ^b^	33.5 ^a^
Mean	18.3 ^f^	29.2 ^d^	33.1 ^c^	25.2 ^e^	42.6 ^b^	33.7 ^c^	26.3 ^e^	46.7 ^a^	
**Shoot dry weight (g)**									
Clay	3.1 ± 0.1 ^i^	6.4 ± 0.6 ^fgh^	6.6 ± 0.3 ^efgh^	5.8 ± 0.5 ^gh^	10.3 ± 0.8 ^ab^	9.7 ± 0.3 ^abc^	6.2 ± 0.4 ^gh^	10.3 ± 1.2 ^ab^	7.3 ^b^
Sand	4.7 ± 0.3 h^i^	7.9 ± 0.8 ^cdefg^	7.3 ± 0.3 ^defg^	8.9 ± 0.4 ^bcd^	10.8 ± 0.4 ^a^	8.6 ± 0.9 ^bcdef^	8.8 ± 0.2 ^bcde^	10.7 ± 0.7 ^a^	8.4 ^a^
Mean	3.9 ^d^	7.1 ^c^	6.9 ^c^	7.4 ^c^	10.5 ^a^	9.2 ^b^	7.5 ^c^	10.5 ^a^	
**Root fresh weight (g)**									
Clay	0.30 ± 0.01 ^j^	0.65 ± 0.01 ^ef^	0.53 ± 0.01 ^g^	0.38 ± 0.01 ^i^	0.70 ± 0.01 ^d^	1.35 ± 0.02 ^a^	0.52 ± 0.01 ^g^	0.82 ± 0.01 ^c^	0.66 ^a^
Sand	0.17 ± 0.00 ^k^	0.44 ± 0.01 ^h^	0.49 ± 0.01 ^g^	0.32 ± 0.01 ^j^	0.98 ± 0.01 ^b^	0.70 ± 0.00 ^df^	0.64 ± 0.01 ^f^	0.79 ± 0.02 ^c^	0.57 ^b^
Mean	0.23 ^h^	0.4 ^e^	0.51 ^f^	0.34 ^g^	0.84 ^b^	1.03 ^a^	0.58 ^d^	0.80 ^c^	
**Root dry weight (g)**									
Clay	0.13 ± 0.00 ^g^	0.25 ± 0.01 ^c^	0.17 ± 0.00 ^e^	0.15 ± 0.01 ^ef^	0.25 ± 0.00 ^c^	0.29 ± 0.01 ^a^	0.20 ± 0.00 ^d^	0.27 ± 0.00 ^b^	0.21 ^a^
Sand	0.06 ± 0.00 ^h^	0.16 ± 0.01 ^e^	0.20 ± 0.00 ^d^	0.14 ± 0.00 ^fg^	0.27 ± 0.01 ^ab^	0.25 ± 0.00 ^c^	0.23 ± 0.01 ^c^	0.29 ± 0.01 ^ab^	0.20 ^b^
Mean	0.09 ^f^	0.20 ^c^	0.19 ^d^	0.15 ^e^	0.26 ^b^	0.27 ^ab^	0.22 ^c^	0.28 ^a^	

CK: control; A: *A. chroococcum*; B: *B. megaterium*; C: *B. cerkularice*. AB, AC, BC, and ABC are microbial consortiums of the PGP bacteria. Different superscript letters indicate statistically significant differences among the soil type, various bacteria and their interactions. Each value represents an average value of three replicates ± SD and averages were compared using LSD at *p* ≤ 0.05.

**Table 3 plants-12-03931-t003:** No. of buds, buds fresh weight (g), buds dry weight (g) and seed dry weight (g) of pea (*Pisum sativum* L.) under the effect of single and consortium plant growth-promoting bacteria (*Azotobacter chroococcum*, *Bacillus megaterium* and *Bacillus cerkularice*) in clay and new reclaimed sandy soil.

**Soil**	**Ck**	**A**	**B**	**C**	**AB**	**AC**	**BC**	**ABC**	**Mean**
	**No. of Buds**								
Clay	11.7 ± 1.2 ^i^	20.0 ± 1.7 ^fgh^	17.7 ± 1.2 ^gh^	16.7 ± 1.5 ^h^	26.7 ± 0.6 ^cd^	23.7 ± 1.5 ^def^	20.3 ± 1.5 ^fgh^	32.0 ± 1.0 ^a^	21.1 ^b^
Sand	19.0 ± 1.0 ^gh^	26.0 ± 1.0 ^d^	31.0 ± 1.0 ^ab^	21.7 ± 1.5 ^efg^	32.3 ± 1.5 ^a^	27.7 ± 1.5 ^bcd^	25.7 ± 0.6 ^de^	30.7 ± 2.1 ^abc^	26.8 ^a^
Mean	15.3 ^e^	23.0 ^c^	24.3 b^c^	19.2 ^d^	29.5 ^a^	25.7 ^b^	23.0 ^c^	31.3 ^a^	
**Bud fresh weight (g)**									
Clay	35.6 ± 0.6 ^i^	60.5 ± 1.0 ^fg^	49.2 ± 1.7 ^h^	39.7 ± 1.6 ^i^	84.1 ± 2.3 ^cd^	53.9 ± 1.5 ^gh^	41.7 ± 0.8 ^i^	77.8 ± 1.7 ^d^	55.3 ^b^
Sand	50.1 ± 1.1 ^h^	64.8 ± 4.0 ^ef^	89.6 ± 2.4 ^bc^	68.4 ± 2.3 ^e^	113.2 ± 4.8 ^a^	70.2 ± 1.2 ^e^	86.5 ± 2.4 ^c^	95.1 ± 2.7 ^b^	79.7 ^a^
Mean	42.9 ^f^	62.7 ^d^	69.4 ^c^	54.0 ^e^	98.7 ^a^	62.0 ^d^	64.1 ^d^	86.5 ^b^	
**Bud dry weight (g)**									
Clay	2.17 ± 0.10 ^i^	4.10 ± 0.17 ^d^	2.83 ± 0.15 ^gh^	2.97 ± 0.15 ^gh^	6.20 ± 0.17 ^a^	4.23 ± 0.15 ^cd^	3.27 ± 0.15 ^fg^	5.53 ± 0.23 ^b^	3.91 ^a^
Sand	3.07 ± 0.12 ^fgh^	3.80 ± 0.10 ^de^	3.77 ± 0.15 ^de^	2.67 ± 0.12 ^h^	4.60 ± 0.26 ^c^	4.07 ± 0.15 ^d^	3.50 ± 0.10 ^ef^	4.26 ± 0.05 ^cd^	3.72 ^b^
Mean	2.62 ^e^	3.95 ^c^	3.30 ^d^	2.82 ^e^	5.40 ^a^	4.15 ^c^	3.38 ^d^	4.90 ^b^	
**Seed dry weight (g)**									
Clay	2.77 ± 0.06 ^i^	8.60 ± 0.26 ^de^	7.43 ± 0.15 ^fg^	6.83 ± 0.21 ^gh^	9.93 ± 0.23 ^ab^	7.20 ± 0.10 ^fgh^	6.77 ± 0.15 ^h^	8.90 ± 0.20 ^cde^	7.30 ^b^
Sand	6.93 ± 0.12 ^gh^	8.20 ± 0.10 ^ef^	9.27 ± 0.46 ^bcd^	7.13 ± 0.21 ^gh^	10.50 ± 0.17 ^a^	8.07 ± 0.15 ^ef^	9.50 ± 0.26 ^bcd^	9.70 ± 0.30 ^bc^	8.66 ^a^
Mean	4.85 ^f^	8.40 ^c^	8.35 ^c^	6.98 ^e^	10.22 ^a^	7.63 ^d^	8.13 ^c^	9.30 ^b^	

CK: control; A: *A. chroococcum*; B: *B. megaterium*; C: *B. cerkularice*. AB, AC, BC, and ABC are microbial consortiums of the PGP bacteria. Different superscript letters indicate statistically significant differences among the soil type, various bacteria and their interactions. Each value represents an average value of three replicates ± SD and averages were compared using LSD at *p* ≤ 0.05.

**Table 4 plants-12-03931-t004:** Physical and chemical properties of the two type soils used in the current investigation.

Soil Texture	Clay Loam	Sandy
Field capacity (%)	40.4	11
Saturation percentage (%)	74.1	24.7
Total CaCO_3_ (g kg^−1^ soil)	32	240
Organic matter (g kg^−1^ soil)	12	3.4
EC (dS m^−1^)	1.25	1.24
pH (1:2.5 water suspension)	7.95	8.2
Soluble cations (mmol_c_ L^−1^):		
Ca^++^	3.2	5.06
Mg^++^	1.3	4.33
Na^+^	7.75	2.22
K^+^	0.25	0.82
Soluble anions (mmol_c_ L^−1^):		
CO_3_^−^ HCO_3_^−^	3.4	3.3
Cl^−^	3	5.24
SO_4_^−^	6.6	3.89
Macronutrients (mg kg^−1^ soil):		
Total N	135	120
Available P	8.2	5.2
Available K	355	50.3

## Data Availability

The data presented in this study are available upon request from the corresponding author.

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
