# Peer review of "Responses of Pea (Pisum sativum L.) to Single and Consortium Bio-Fertilizers in Clay and Newly Reclaimed Soils"

_plants, 2023, doi:10.3390/plants12233931_

Round 1
Reviewer 1 Report
Comments and Suggestions for Authors
The authors must seek the help of a scientific editor to help them produce a viable manuscript for publication. In addition, there is a need to provide a more nuanced introduction and justification of the study, with clear objectives. The materials and methods are sketchy and do not meet the normal standards of repeatability i.e., providing such detail and clarity that other researchers can repeat the study and validate the results of this study or otherwise.
The "Results and Discussion" section is too long and tiresome to read. Some of the variables follow a similar pattern, which could be taken into account to make the description of the results shorter and more effective. Also, the discussion can better integrate these results to become more straightforward. That could be achieved if the description of results is separated from the Discussion section.
Also make correlation plots i.e., pearson correlation or cluster plot with convex hull to elaborate the data matrix correlation. It will increase the quality and value of data explained.
or this. Improved the conclusion and provide conclusive statements on the basis of results authors have observed.
• Give future recommendations as well.
• Provide a statement showing potential benefits for growers.
The scientific background of the topic is poor. In "Introduction" and "Discussion", the authors should cite recent references between 2016-2023 from JCR journals.
Abdul S, Muhammad AA, Shabir H, Hesham A. El E, Sajjad H, Niaz A, Abdul G, RZ Sayyed , Fahad S, Subhan D, Rahul D (2021a) Zinc nutrition and arbuscular mycorrhizal symbiosis effects on maize (Zea mays L.) growth and productivity. J Saudi Society Agricultural Sci https://doi.org/10.1016/j.sjbs.2021.06.096
Abdul S, Muhammad AA, Subhan D, Niaz A, Fahad S, Rahul D, Mohammad JA, Omaima N, Muhammad Habib ur R, Bernard RG (2021b) Effect of arbuscular mycorrhizal fungi on the physiological functioning of maize under zinc‑deficient soils. Sci Rep 11:18468 https://doi.org/10.1038/s41598-021-97742-1
Comments on the Quality of English LanguageSee the above comments
Author Response
Thank you for your letter and for the reviewers’ comments concerning our manuscript. Those comments are all valuable and very helpful for revising and improving our paper, as well as the important guiding significance to our research. We have studied the comments carefully and made corrections that we hope meet with approval. Revised portions are marked in red on the paper. The main corrections in the paper and the responses to the reviewer’s comments are as flowing:
The authors must seek the help of a scientific editor to help them produce a viable manuscript for publication. In addition, there is a need to provide a more nuanced introduction and justification of the study, with clear objectives. The materials and methods are sketchy and do not meet the normal standards of repeatability i.e., providing such detail and clarity that other researchers can repeat the study and validate the results of this study or otherwise.
Response: We thank you for your valuable comments, the manuscript was English edited and revised. We have revised and modified the introduction part with more recent references and a clear justification of the study. The method part in the revised manuscript was modified with more information’s, and a diagram depicting the experiment design was added and mentioned as “Summarization of the harvest intervals and data collected throughout the study from transplanting to finalizing” is provided in Fig. 9.
The "Results and Discussion" section is too long and tiresome to read. Some of the variables follow a similar pattern, which could be taken into account to make the description of the results shorter and more effective. Also, the discussion can better integrate these results to become more straightforward. That could be achieved if the description of results is separated from the Discussion section.
Response: We thank you for your valuable comments, we decreased the length of the result as much as possible and modified the discussion part in the revised manuscript.
Also make correlation plots i.e., pearson correlation or cluster plot with convex hull to elaborate the data matrix correlation. It will increase the quality and value of data explained.
or this. Improved the conclusion and provide conclusive statements on the basis of results authors have observed.
- Give future recommendations as well.
- Provide a statement showing potential benefits for growers.
Response: We really thank you for your valuable comments and we added cluster plots to the results section to increase the quality and value of data explained as recommended. We also modified the conclusion part and added future recommendations.
The scientific background of the topic is poor. In "Introduction" and "Discussion", the authors should cite recent references between 2016-2023 from JCR journals.
Abdul S, Muhammad AA, Shabir H, Hesham A. El E, Sajjad H, Niaz A, Abdul G, RZ Sayyed , Fahad S, Subhan D, Rahul D (2021a) Zinc nutrition and arbuscular mycorrhizal symbiosis effects on maize (Zea mays L.) growth and productivity. J Saudi Society Agricultural Sci https://doi.org/10.1016/j.sjbs.2021.06.096
Abdul S, Muhammad AA, Subhan D, Niaz A, Fahad S, Rahul D, Mohammad JA, Omaima N, Muhammad Habib ur R, Bernard RG (2021b) Effect of arbuscular mycorrhizal fungi on the physiological functioning of maize under zinc‑deficient soils. Sci Rep 11:18468 https://doi.org/10.1038/s41598-021-97742-1
Response: We appreciate the reviewer's comments regarding the improvement of our Ms. We revised the introduction and discussion parts and included only recent references between 2016-2023 from JCR journals as recommended. We have also incorporated all these references.
- The State of Food and Agriculture 2016. Climate Change, Agriculture and Food Security; FAO: Rome, Italy, 2016.
- Demir, H.; Sönmez, Ë™I.; Uçan, U.; Akgün, Ë™I.H. Biofertilizers Improve the Plant Growth, Yield, and Mineral Concentration of Lettuce and Broccoli. Agronomy 2023, 13, 2031. https://doi.org/10.3390/
- Powers, S.E.; Thavarajah, D. Checking Agriculture’s Pulse: Field Pea (Pisum Sativum), Sustainability, and Phosphorus Use Efficiency. Front. Plant Sci. 2019, 10, 1489. https://doi.org/10.3389/fpls.2019.01489
- Sun, C.X.; Bei, K.; Liu, Y.H.; Pan, Z.Y. Humic Acid Improves Greenhouse Tomato Quality and Bacterial Richness in Rhizosphere Soil. ACS Omega 2022, 7, 29823–29831. https://doi.org/1021/acsomega.2c02663
- de Andrade, L.A.; Santos, C.H.B.; Frezarin, E.T.; Sales, L.R.; Rigobelo, E.C. Plant Growth-Promoting Rhizobacteria for Sustainable Agricultural Production. Microorganisms 2023, 11, 1088. https://doi.org/10.3390/microorganisms11041088.
- Chamekh, A.; Kharbech, O.; Fersi, C.; Driss Limam, R.; Brandt, K.K.; Djebali, W. Chouari R. Insights on strain 115 plant growth-promoting bacteria traits and its contribution in lead stress alleviation in pea (Pisum sativum) plants. Arch Microbiol. 2022, 27, 205. https://doi.org/10.1007/s00203-022-03341-7.
- Shabaan, M.; Asghar, H.N.; Akhtar, M.J.; Ali, Q.; Ejaz, M. Role of plant growth promoting rhizobacteria in the alleviation of lead toxicity to Pisum sativum Int J Phytoremediation 2021, 23,837-845. https://doi.org/10.1080/15226514.2020.1859988.
- Ramakrishna W.; Yadav, R.; Li, K. Plant growth promoting bacteria in agriculture: Two sides of a coin. Soil Eco., 2019 138:10-18. https://doi.org/10.1016/j.apsoil.2019.02.019.
- Bungau, S.; Behl, T.; Aleya, L.; Bourgeade, P.; Aloui-Sossé, B.; Purza, A.L.; Abid, A.; Samuel, A.D. Expatiating the impact of anthropogenic aspects and climatic factors on long-term soil monitoring and management. Environ Sci Pollut Res 2021, 28, 30528–30550. https://doi.org/10.1007/s11356-021-14127-7.
- Bizos, G.; Papatheodorou, E.M.; Chatzistathis, T.; Ntalli, N.; Aschonitis, V.G.; Monokrousos, N. The Role of Microbial Inoculants on Plant Protection, Growth Stimulation, and Crop Productivity of the Olive Tree (Olea europea). Plants 2020, 9, 743. https://doi.org/10.3390/plants9060743.
- Wang, L.; Yang, F.; Yaoyao, E.; Yuan, J.; Raza, W.; Huang, Q.; Shen, Q. Long-term application of bioorganic fertilizers improved soil biochemical properties and microbial communities of an apple orchard soil. Front Microbiol 2016, 7, 1893. https://doi.org/3389/fmicb.2016.01893
- Saboor, A.; Ali, M.A.; Danish, S.; Ahmed, N.; Fahad, S.; Datta, R.; Ansari, M.J.; Nasif, O.; Rahman, M.H.U.; Glick, B.R. Effect of arbuscular mycorrhizal fungi on the physiological functioning of maize under zinc-deficient soils. Rep. 2021a, 11, 18468. https://doi.org/10.1038/s41598-021-97742-1.
- Saboor A., Ali M.A., Hussain S., El Enshasy H.A., Hussain S., Ahmed N., Gafur A., Sayyed R., Fahad S., Danish S. Zinc nutrition and arbuscular mycorrhizal symbiosis effects on maize (Zea mays) growth and productivity. Saudi J. Biol. Sci. 2021b; 28:6339–6351. https://doi.org/10.1016/j.sjbs.2021.06.096.
- Akhtar, M.; Yousaf, S.;Sarwar, N.; Hussain, S.. Zinc biofortification of cereals— role of phosphorus and other impediments in alkaline calcareous soils. Geochem. Health 2019, 41, 2365–2379. https://doi.org/10.1007/s10653-019-00279-6.
- Tahir, S.; Marschner, P. Clay Addition to Sandy Soil—Influence of Clay Type and Size on Nutrient Availability in Sandy Soils Amended with Residues Differing in C/N ratio, Pedosphere 2017, 27, 293-305. https://doi.org/10.1016/S1002-0160(17)60317-5.
- Gu, X.; Liu, Y.; Li, N.; Liu, Y.; Zhao, D.; Wei, B.; Wen, X. Effects of the Foliar Application of Potassium Fertilizer on the Grain Protein and Dough Quality of Wheat. Agronomy 2021, 11, 1749. https://doi.org/3390/agronomy11091749.
- Trovato, M.; Funck, D.; Forlani, G.; Okumoto, S.; Amir, R. Amino Acids in Plants: Regulation and Functions in Development and Stress Defense. Plant Sci. 2021, 12, 772810. https://doi.org/10.3389/fpls.2021.772810.
- Samuel, A.D.; Bungau, S.; Fodor I.K., Tit, D.M.; Blidar, C.F.; David, A.T.; Melinte, C.E. Effects of Liming and Fertilization on the Dehydrogenase and Catalase Activities. Chim. 2019, 70(10), 3464-3468. https://doi.org/10.37358/RC.19.10.7576
- Abeed, A. H.; AL-Huqail, A. A.; Albalawi, S.; Alghamdi, S. A.; Ali, B.; Alghanem, S. M.; El-Mahdy, M. T. Calcium nanoparticles mitigate severe salt stress in Solanum lycopersicon by instigating the antioxidant defense system and renovating the protein profile. South Afri J Bot 2023, 161, 36-52. https://doi.org/10.1016/j.sajb.2023.08.005
- Abeed, A.H.A.; Salama, F.M. Attenuating Effect of an Extract of Cd-Hyperaccumulator Solanum nigrum on the Growth and Physio-chemical Changes of Datura innoxia Under Cd Stress. J Soil Sci Plant Nutr 2022, 22. https://doi.org/10.1007/s42729-022-00966-x.
Reviewer 2 Report
Comments and Suggestions for Authors
The authors focused on the Responses of pea (Pisum sativum L.) to single and consortium bio-fertilizers in clay and newly reclaimed soils. Good results part, but incomplete Discussion. Please see my suggestions bellow:
Final of Introduction. It must be addressed from the perspective of describing the contribution to the field under analysis and the elements of scientific novelty presented, as the LAST, SEPARATE paragraph of this section, to be easier visible. Develop it better. What differentiate your paper from other in the same topic? Give a reason for interest in this paper.
Check the entire manuscript. l, ml, etc. should be corrected as L, mL, etc., Litter being the international unit of measure for volume. Be consistent in denotation!
Discussion section is poor and short, it should be completed for a better framing of the topic as follows:
- How it can be improved the pea crop management under the effects of climate change? I suggest checking and referring to and https://doi.org/10.1007/s11356-021-14127-7
- Regarding the effects of long-term application of organic and mineral fertilizers on soil enzymes they must be detailed, you should check https://doi.org/10.37358/RC.19.10.7576 https://doi.org/10.37358/RC.18.10.6590
- Also, the importance of nanotechnology was omitted; it should be provided in a paragraph, regarding the aspect of nano-farming versus nanotoxicity – please see https://doi.org/10.1016/j.chemosphere.2021.132533
- Additionally, a last paragraph of Discussion section should be added, describing the Strengths and the Weakness/limitations of your research/results.
Please provide complete information in the 4th section:
- the Model, Producer/manufacturer, City, and Country for EACH APPARATUS (4 information) used in the research, and
- the Producer, Country, purity degree, and concentration or CAS (4 information) used for EACH REAGENT/chemical used. Check the entire manuscript in this regard. This information gives the possibility for replicating you experiment to other authors and are requested in ALL journals.
Comments on the Quality of English LanguageIt should be revised
Author Response
Thank you for your letter and for the reviewers’ comments concerning our manuscript. Those comments are all valuable and very helpful for revising and improving our paper, as well as the important guiding significance to our research. We have studied the comments carefully and made corrections that we hope meet with approval. Revised portions are marked in red on the paper. The main corrections in the paper and the responses to the reviewer’s comments are as flowing:
The authors focused on the Responses of pea (Pisum sativum L.) to single and consortium bio-fertilizers in clay and newly reclaimed soils. Good results part, but incomplete Discussion. Please see my suggestions bellow:
Final of Introduction. It must be addressed from the perspective of describing the contribution to the field under analysis and the elements of scientific novelty presented, as the LAST, SEPARATE paragraph of this section, to be easier visible. Develop it better. What differentiate your paper from other in the same topic? Give a reason for interest in this paper.
Response: We really thank you for your valuable comments and we revised the introduction part and included recent references and modified the objective paragraph as recommended.
Check the entire manuscript. l, ml, etc. should be corrected as L, mL, etc., Litter being the international unit of measure for volume. Be consistent in denotation!
Response: We thank you for your valuable comments and we have corrected it in the revised manuscript.
Discussion section is poor and short, it should be completed for a better framing of the topic as follows:
- How it can be improved the pea crop management under the effects of climate change? I suggest checking and referring to and https://doi.org/10.1007/s11356-021-14127-7
- Regarding the effects of long-term application of organic and mineral fertilizers on soil enzymes they must be detailed, you should check https://doi.org/10.37358/RC.19.10.7576 https://doi.org/10.37358/RC.18.10.6590
- Also, the importance of nanotechnology was omitted; it should be provided in a paragraph, regarding the aspect of nano-farming versus nanotoxicity – please see https://doi.org/10.1016/j.chemosphere.2021.132533ز
Response: We really thank you for your valuable comments and we revised the discussion part and included these significant studies and other recent references as recommended. We have also incorporated all these references.
- Bungau, S.; Behl, T.; Aleya, L.; Bourgeade, P.; Aloui-Sossé, B.; Purza, A.L.; Abid, A.; Samuel, A.D. Expatiating the impact of anthropogenic aspects and climatic factors on long-term soil monitoring and management. Environ Sci Pollut Res 2021, 28, 30528–30550. https://doi.org/10.1007/s11356-021-14127-7.
- Bizos, G.; Papatheodorou, E.M.; Chatzistathis, T.; Ntalli, N.; Aschonitis, V.G.; Monokrousos, N. The Role of Microbial Inoculants on Plant Protection, Growth Stimulation, and Crop Productivity of the Olive Tree (Olea europea). Plants 2020, 9, 743. https://doi.org/10.3390/plants9060743.
- Wang, L.; Yang, F.; Yaoyao, E.; Yuan, J.; Raza, W.; Huang, Q.; Shen, Q. Long-term application of bioorganic fertilizers improved soil biochemical properties and microbial communities of an apple orchard soil. Front Microbiol 2016, 7, 1893. https://doi.org/3389/fmicb.2016.01893
- Saboor, A.; Ali, M.A.; Danish, S.; Ahmed, N.; Fahad, S.; Datta, R.; Ansari, M.J.; Nasif, O.; Rahman, M.H.U.; Glick, B.R. Effect of arbuscular mycorrhizal fungi on the physiological functioning of maize under zinc-deficient soils. Rep. 2021a, 11, 18468. https://doi.org/10.1038/s41598-021-97742-1.
- Saboor A., Ali M.A., Hussain S., El Enshasy H.A., Hussain S., Ahmed N., Gafur A., Sayyed R., Fahad S., Danish S. Zinc nutrition and arbuscular mycorrhizal symbiosis effects on maize (Zea mays) growth and productivity. Saudi J. Biol. Sci. 2021b; 28:6339–6351. https://doi.org/10.1016/j.sjbs.2021.06.096.
- Akhtar, M.; Yousaf, S.;Sarwar, N.; Hussain, S.. Zinc biofortification of cereals— role of phosphorus and other impediments in alkaline calcareous soils. Geochem. Health 2019, 41, 2365–2379. https://doi.org/10.1007/s10653-019-00279-6.
- Tahir, S.; Marschner, P. Clay Addition to Sandy Soil—Influence of Clay Type and Size on Nutrient Availability in Sandy Soils Amended with Residues Differing in C/N ratio, Pedosphere 2017, 27, 293-305. https://doi.org/10.1016/S1002-0160(17)60317-5.
- Gu, X.; Liu, Y.; Li, N.; Liu, Y.; Zhao, D.; Wei, B.; Wen, X. Effects of the Foliar Application of Potassium Fertilizer on the Grain Protein and Dough Quality of Wheat. Agronomy 2021, 11, 1749. https://doi.org/3390/agronomy11091749.
- Trovato, M.; Funck, D.; Forlani, G.; Okumoto, S.; Amir, R. Amino Acids in Plants: Regulation and Functions in Development and Stress Defense. Plant Sci. 2021, 12, 772810. https://doi.org/10.3389/fpls.2021.772810.
- Samuel, A.D.; Bungau, S.; Fodor I.K., Tit, D.M.; Blidar, C.F.; David, A.T.; Melinte, C.E. Effects of Liming and Fertilization on the Dehydrogenase and Catalase Activities. Chim. 2019, 70(10), 3464-3468. https://doi.org/10.37358/RC.19.10.7576
- Abeed, A. H.; AL-Huqail, A. A.; Albalawi, S.; Alghamdi, S. A.; Ali, B.; Alghanem, S. M.; El-Mahdy, M. T. Calcium nanoparticles mitigate severe salt stress in Solanum lycopersicon by instigating the antioxidant defense system and renovating the protein profile. South Afri J Bot 2023, 161, 36-52. https://doi.org/10.1016/j.sajb.2023.08.005
- Abeed, A.H.A.; Salama, F.M. Attenuating Effect of an Extract of Cd-Hyperaccumulator Solanum nigrum on the Growth and Physio-chemical Changes of Datura innoxia Under Cd Stress. J Soil Sci Plant Nutr 2022, 22. https://doi.org/10.1007/s42729-022-00966-x.
- Additionally, a last paragraph of Discussion section should be added, describing the Strengths and the Weakness/limitations of your research/results.
Response: We really thank you for your valuable comments and we included this part as a conclusion in the revised manuscript.
Please provide complete information in the 4th section:
- the Model, Producer/manufacturer, City, and Country for EACH APPARATUS (4 information) used in the research, and
- the Producer, Country, purity degree, and concentration or CAS (4 information) used for EACH REAGENT/chemical used. Check the entire manuscript in this regard. This information gives the possibility for replicating you experiment to other authors and are requested in ALL journals.
Response: We appreciate the reviewer's comments regarding the improvement of the materials and methods section. We have corrected the method part and added all the requested information in the revised manuscript.
Round 2
Reviewer 1 Report
Comments and Suggestions for Authors
Accepted
Comments on the Quality of English LanguageN.A
Reviewer 2 Report
Comments and Suggestions for Authors
The authors responded to my requests.
Comments on the Quality of English LanguageMinor Editing